# Mass Balance and Compositional Analysis of Biomass Outputs from Cacao Fruits

**DOI:** 10.3390/molecules27123717

**Published:** 2022-06-09

**Authors:** Marisol Vergara-Mendoza, Genny R. Martínez, Cristian Blanco-Tirado, Marianny Y. Combariza

**Affiliations:** Escuela de Química, Universidad Industrial de Santander, Bucaramanga 680002, Colombia; marisolvergara14@yahoo.es (M.V.-M.); genmarba@gmail.com (G.R.M.); cblancot@uis.edu.co (C.B.-T.)

**Keywords:** cacao fruit, cacao husk, cacao mucilage exudate, cacao placenta, cacao beans, valorization

## Abstract

The global chocolate value chain is based exclusively on cacao beans (CBs). With few exceptions, most CBs traded worldwide are produced under a linear economy model, where only 8 to 10% of the biomass ends up in chocolate-related products. This contribution reports the mass balance and composition dynamics of cacao fruit biomass outputs throughout one full year of the crop cycle. This information is relevant because future biorefinery developments and the efficient use of cacao fruits will depend on reliable, robust, and time-dependent compositional and mass balance data. Cacao husk (CH), beans (CBs), and placenta (CP) constitute, as dry weight, 8.92 ± 0.90 wt %, 8.87 ± 0.52 wt %, and 0.57 ± 0.05 wt % of the cacao fruit, respectively, while moisture makes up most of the biomass weight (71.6 ± 2.29 wt %). CH and CP are solid lignocellulosic outputs. Interestingly, the highest cellulose and lignin contents in CH coincide with cacao’s primary harvest season (October to January). CB contains carbohydrates, fats, protein, ash, and phenolic compounds. The total polyphenol content in CBs is time-dependent, reaching maxima values during the harvest seasons. In addition, the fruit contains 4.13 ± 0.80 wt % of CME, a sugar- and nutrient-rich liquid output, with an average of 20 wt % of simple sugars (glucose, fructose, and sucrose), in addition to minerals (mainly K and Ca) and proteins. The total carbohydrate content in CME changes dramatically throughout the year, with a minimum of 10 wt % from August to January and a maximum of 29 wt % in March.

## 1. Introduction

Cacao is an important crop in the equatorial regions of Latin America, Western Africa, and Southeast Asia, providing income to more than 4.5 million families worldwide. According to the International Cocoa Organization (ICCO), world gross cacao bean (CB) production in 2019 was 4.75 × 10^6^ tons [1]. The main cacao-producing countries are Côte d’Ivoire, Ghana, Indonesia, Brazil, Cameroon, Nigeria, Ecuador, Peru, and the Dominican Republic. In South America, more than 90% of CBs come from Brazil, Ecuador, Peru, Colombia, and Mexico [2]. In 2019, Colombia produced 5.97 × 10^4^ tons of CBs, an increase of 4.9% with respect to 2018. The Santander region in Colombia contributes 42% of the country´s CBs [3]. As with many other agricultural commodities, CB production generates abundant residual biomass. Typically, only 8 to 10% of the biomass contained in the cacao fruit ends up as cacao beans, while the remaining biomass is discarded as waste. Figure 1 shows the traditional approach to cacao fruit processing worldwide. Clearly, the current process is aimed (with a few exceptions) exclusively at obtaining fermented and dry cacao beans. Considering the scheme in Figure 1 as the standard process for cacao fruit, the residual biomass from cacao crops worldwide could easily reach up to 4.5 × 10^7^ tonnes/year (2019).

The cacao fruit (Figure 1) has a dense shell, or cacao husk (CH), protecting the seeds. Once harvested, the fruit is opened, the husk is discarded, and the seeds (covered by a white mucilage) are stored and fermented in heaps or wooden crates to produce cacao beans. Literature reports from 2000 to 2021 [4,5,6,7] show a steady increase in the number of publications related to the use of some residual biomass outputs from cacao bean production. The scientific interest in residual biomass usage is fueled by the need to increase the cacao crop’s circularity, strengthen the cacao value chain, increase the economic profitability of the crop for producers, and reduce the environmental impacts.

There are many uses for cacao residual biomass, particularly cacao husk (CH), involving energy generation, biomaterials extraction, the isolation of active ingredients, catalyst support, human food, animal feed, and biocomposite synthesis, among others. For instance, cacao husk (CH) anaerobic digestion and fermentation result in biogas production [8,9,10]. Energetic potential analysis showed that gasification is better than pyrolysis or combustion for CH processing [11]. Along the same lines, the CH potential for electricity generation in Uganda was evaluated, reporting a potential energy generation of 24 × 10^6^ MWh for the 9.03 × 10^3^ tonnes of CH produced in Uganda in 2018 [12].

Materials production is another alternative for utilizing cacao residual biomass. For instance, controlled CH pyrolysis can be used for activated carbon production. Some authors reported that the yields and textural properties of the material improve by using acid (H_3_PO_4_) or basic (KOH) treatments [13]. An activated carbon surface area of over 1300 m^2^/g using de-ashed CH as the raw material has been reported [14]. Activated carbon derived from CH exhibited an adsorption performance comparable to that of commercial activated carbon for the removal of Congo Red and residual drugs (diclofenac) from aqueous solutions [15,16]. Similarly, there are reports of catalyst and nanoparticle synthesis using CH; for example, the production of Neem seed oil methyl esters, using CH ash as a catalyst, or silver nanoparticle synthesis using CH extract [17,18]. Additionally, researchers reported the use of the pectins in CH as a nutraceutical and functional pharmaceutical excipients [19]. CH has also been tested as a filler in a bioplastic-based composite [20] and a synthetic PP composite [21]. The tensile strength, flexural strength, and modulus of thermoplastic polyurethane (TPU) were improved by the addition of CH fibers [22]. In the field of animal nutrition, CH can be a substitute for traditional ingredients, like replacing maize as a dietary staple for raising rabbits [23] or pigs [24].

However, there is little information in the literature regarding the uses of biomass outputs from cacao fruit other than CH, such as cacao placenta (CP) or cacao mucilage exudate/sweatings (CME). Cacao mucilage exudate (CME), in particular, is generally lost as a lixiviate during cacao bean fermentation. However, it is occasionally consumed as a fresh drink or is used for manufacturing syrups, jams, marmalades, and alcoholic beverages [25,26,27,28,29,30,31,32]. Likewise, the technological applications of CME include ethanol production by fermentation [33]. Recently, our group reported the use of CME as culture media for bacterial cellulose biosynthesis [34]. Finally, the cacao placenta (CP), a tissue that holds the cacao beans together inside the pod (Figure 1), has been used in the production of beverages and nutritional bars [35].

Cacao bean prices, like any other commodity in the world, are controlled globally by the interaction between supply and demand chains. However, global markets are changing, and chocolate consumers are driven by additional factors such as fair trade, direct trade, and reduced environmental impacts (directly associated with crop circularity and the emission of pollutants during beans processing). The cacao fruit has a lot to offer besides its beans, and with cacao production and consumption on the rise, there is abundant residual biomass from the crop that can be used to strengthen the crop value chain, particularly in the producers’ countries. However, despite the many possible uses for residual biomass from cacao, currently, there is a systematic lack of information in terms of compositional and mass balance data for the whole cacao fruit, let alone data regarding the seasonal variations in these fundamental crop characteristics. Most of the compositional data available in the current literature relates to small samples of cacao fruit collected exclusively during the harvest months. Knowing the seasonal compositional and mass balance changes in biomass outputs from cacao fruit processing is of fundamental importance to planning for efficient usage of the residual biomass from fermented beans production. In addition, future biorefinery approaches and final usage methods for cacao fruit biomass will be determined by its abundance and composition.

Thus, we hypothesize that it is possible to observe cacao fruit composition and mass-balance dynamics by analyzing representative one-tonne samples, collected monthly for one year. Initially, we separated the traditional biomass outputs of cacao fruit processing (as seen in Figure 1) and determined their abundance, as percentages by weight of the fruit. Next, we collected, processed, and stored all cacao fruit biomass outputs for a detailed mass balance and compositional analysis. CH, CP, CBs, and a new biomass output labeled cacao mucilage exudate (CME) were studied. In the traditional cacao-processing scheme, only a low percentage of the cacao fruit biomass, represented in dry/fermented CBs, is used in the chocolate industry. The remaining percentage, represented in dry CH and CP, and CME are considered to be residual biomass. These residues could be used as raw materials to feed other processes to produce value-added products, to strengthen the cacao value chain in the producers’ countries.

## 2. Results and Discussion

### 2.1. Cacao Fruit Mass Balance

The traditional process for cacao bean production is manually intensive and starts with cacao fruit harvesting, followed by fruit-opening and the extraction of the fresh cacao beans (CBs), as shown in Figure 1. All these processes are performed on-site (on-farm). Typically, the empty fruit husk (CH) and the placenta (CP) are discarded and left to decompose in the field. In the traditional process, only fresh CBs undergo further treatment, in a step called fermentation.

Figure 2 shows the seasonal percentage by weight variation of cacao beans (CBs), cacao placenta (CP), and cacao fruit husk (CH), of twelve cacao fruit loads (one tonne each), gathered over the months of February to January, using the traditional cacao fruit-processing scheme (Figure 1). Appendix A contains detailed wt % data for fresh CH, CP, and CBs in every cacao-fruit load processed. The percentage by weight values reported in Figure 2 correspond to the fresh (wet) by-products with respect to the fresh fruit weight. Fresh CH is by far the most abundant cacao fruit byproduct, with a total average of 66.96 ± 2.83 wt %, followed by fresh CBs, with 24.55 ± 1.51 wt %, and CP, with 2.58 ± 0.22 wt %. The losses (5.93 ± 3.15 wt %) correspond to missing CH pieces and spoiled fruits and beans. These percentages are similar to those reported for cacao fruit from Bundibugyo District in Uganda (clone not specified) [8], for cacao fruit samples from Ghana (clone not specified) [36], and for cacao fruit from plantations located in Perak, Malaysia (clone not specified) [37]. The percentage by weight of the fresh by-products did not change significantly for cacao fruit loads gathered over the course of one year, as seen in Figure 2.

For the cacao fruit compositional and mass balance analyses, we used a two-stage procedure shown in Figure 3. The first stage involved opening the cacao fruit and separating the fresh cacao beans (CBs) and the placenta (CP) from the husk (CH), as discussed above. This first process represents the traditional approach to cacao fruit usage (Also shown in Figure 1). In the second stage, the CP and the CH were ground, dried, and stored for further compositional analysis (Figure 3). Once the cacao fruit components were separated and their fresh percentage by weight measured, we subjected the materials to different procedures, as shown in Figure 3 (stage 2). Fresh CBs were fermented for seven days, then the cacao mucilage exudate (CME) lixiviated from the process was collected. Fresh CP and CH were ground and dried. Figure 4 shows the seasonal variation in percentage by weight of dried cacao husk (CH), dried cacao placenta (CP), liquid cacao mucilage exudate (CME), and fermented and dried cacao beans (CBs). Appendix A contains detailed wt % data for dried CH, CP, CBs, and liquid CME in every cacao fruit load processed. Dried CB and CH exhibit similar average percentages by weight of 8.92 ± 0.90 wt % and 8.87 ± 0.52 wt %, respectively (see Appendix A). Dried CP amounts to a small contribution of 0.57 ± 0.05 wt %, as well as CME, with 4.13 ± 0.80 wt %. These results show that four different biomass outputs (dried CH, dried CP, dried CBs, and CME) can be derived from the cacao fruit. Each type of biomass has a particular composition and potential use, as discussed in the following sections.

The mass balance for the first and second stages of cacao fruit processing is shown in Figure 5. The values reported correspond to the actual weights measured while processing the load collected in November (Nov.), the main cacao fruit harvest season in Colombia. Appendix A Information also contains mass balance information for the first and second stages of cacao fruit processing, with averaged wt % data for all biomass outputs. Starting with one tonne of the fresh cacao fruit, we collected 66.3 kg of fresh CH (66.33 wt %), 26.4 kg of fresh CP (2.64 wt %), and 256.9 kg of fresh CBs (25.69 wt %). These materials are high in moisture; after the second stage, which involves grinding (CH, CP), drying (CH, CP), and fermenting (CBs), their mass is reduced dramatically. At this point, 89.4 kg of dried CH, 87.7 kg of dry/fermented CBs, and 5.7 kg of dried CP are the solid materials resulting from the second stage. In addition, a liquid by-product, cacao mucilage exudate CME, can be recovered during the second stage. The CME (43.1 kg) is produced as a lixiviate during the first hours of CB fermentation. Overall, the moisture content in the solid materials (CH, CBs, and CP) present in one tonne of fresh cacao fruit (Nov.) was 720.7 kg. It is worth noting that the traditional way of cacao fruit processing (first stage, Figure 4) results in only 8.77 wt % of useful product, in the form of fermented/dry cacao beans. Ideally, other byproducts of cacao fruit processing, such as CH, CP, and CME, corresponding to an additional 13.82 wt % of the fruit, could be potentially used as raw materials for value-added products. However, the use of these residues depends on availability and the compositional data, which are relatively scarce in the scientific literature. We provide average compositional data for the main biomass outputs, along with compositional changes over one year, in the following sections.

### 2.2. Compositional Analysis

#### 2.2.1. Cacao Fruit Husk (CH)

The cacao fruit husk (CH) functions as a protective pod for the seeds. Thus, CH exhibits a complex chemical structure involving supportive tissue of high mechanical resistance and high water-holding capability. Table 1 shows the average proximate analysis of fresh cacao husk (CH) clone CCN 51, corresponding to twelve loads (one tonne each) of cacao fruit collected over one year. Appendix A also contains detailed wt % CH proximate analysis data for each cacao fruit load processed. The proximate analysis provides a broad classification of components in biomass, which is information of fundamental importance, mainly for animal feed evaluation and biomass energy use. CH is mainly composed of water (84.62 ± 1.97 wt %) and solids (15.38 ± 1.97 wt %). In the literature, the values reported for ash (9.1 wt %) and crude fiber (30.93) [38] are slightly lower than those found in this work (Table 1). Additionally, the reported protein contents of 5.9 wt % [38] are comparable to the 5.44 wt % found in this work. Another study in the literature [39] shows lower values for protein content of 2.42 ± 0.37 wt %, along with 0.93 ± 0.34 for crude fat and 87.06 ± 0.58 moisture, with a comparable value of 10.65 wt % for the ash.

The protein, fat, and crude fiber content in CH makes it attractive as animal feed. For example, ruminants fed with CH from West Java, Indonesia (clone not specified) gained more weight than control ruminants [40]. CH-fed rabbits (20% of their food intake) from Ghana exhibited increased weight compared to corn-fed control animals [9]. In addition, replacing maize with CH (variety not specified) in the diet of *Oreochromis niloticus* (tilapia) reduces its culture costs without significantly affecting the fish’s survival rates and weight [41].

Nowadays, non-edible lignocellulosic materials, readily available as residues from agro-industrial processes, are being considered as alternatives to renewable chemicals and fuels to replace oil-based products. Table 2 compares previous works on cacao husk composition in terms of structural carbohydrates (cellulose and hemicellulose) and lignin, with the average values for twelve cacao loads, as measured in this work. Structural carbohydrates and lignin, collectively, account for up to 78 wt % of the total solids in cacao husk (CH). The average cellulose content for all loads in terms of CH (25.64 ± 3.49 wt %) is lower than that found in various agricultural biomass outputs, such as rice husk (37.1 wt %), sugarcane bagasse (32–44 wt %), cassava peels (37.9 wt %), corn cob (38.27 wt %) and sweet sorghum bagasse (45 wt %). Likewise, the hemicellulose content in CH (19.96 ± 2.42 wt %) is lower than the values found in nutshells and cassava peels, which are 25–30 and 23.9 wt %, respectively. Lignin content, on the other hand, is higher in CH (32.73 ± 5.15 wt %) than that in spent coffee grounds (23.5 wt %), rice husk (24.1 wt %), sugarcane bagasse (19–24 wt %), cassava peels (7.5 wt %), corn cob (7.16 wt %), and sweet sorghum bagasse (21 wt %) [42,43,44,45,46,47]. Appendix A contains detailed compositional information on various agricultural biomasses for comparison purposes.

The average cellulose content in Colombian CH from the CCN 51 clone (25.64 ± 3.49 wt %) is lower than the values reported for CH from Nigeria, in the Ile-Ife region [10]; Brazil, in the Bahia area [48]; Indonesia, in Sumatra province (clone not specified) [50]; and Malaysia, in Jabatan (clone not specified) [42]. The cellulose contents in CH of 41.92 wt % for Peruvian cacao samples (clone not specified) are the highest values found in the literature so far [51]. We found average hemicellulose values of 19.96 ± 2.42 wt % for CCN 51 CH, higher than the values of 10.0, 10.4, and 12 wt % reported for two Colombian and one Ghanaian CH samples [49,50,52]. In contrast, the average hemicellulose values in this work are lower than the 37 ± 0.5 and 35.26 ± 0.05 wt % reported for the Malaysian and Nigerian CH samples [44,51]. On the other hand, the average lignin content of 32.73 ± 5.15 wt % in the CH of clone CCN51 measured in this study is at the high end of the range of values reported by other authors (Table 2). The high lignin content in CCN51 CH could be a characteristic of the clone. However, the literature reporting cacao fruit composition, cited in this work and used to compare our findings, does not include information on the type of material examined. Additionally, to the best of our knowledge, the information currently reported in the literature is derived from grab-sampling of the biomass, understood as samples reflecting the composition at that point in time when the sample was collected. In contrast, in this contribution, we follow changes in cacao fruit composition over time.

The structural biopolymers in CH showed variations over the one-year period considered in this work, as seen in Figure 6. The cellulose contents in CH ranged from 22 to 35 wt % and hemicellulose ranged from 14 to 24 wt %. Lignin content, on the other hand, exhibited significant variability, with values ranging from 25 to 39 wt %. Appendix A of the Supplementary Information contains detailed wt % compositional data for the structural biopolymers and lignin in fresh CH, as fractions of the total solids, for each cacao fruit load. The cellulose, hemicellulose, and lignin contents in cacao fruit can be influenced by factors such as location, tree age, and growth conditions. In Colombia, the cacao crop generally has two harvest seasons per year [2]. The highest productivity season ranges from October to January, while an additional, lower-productivity season runs from April to June. We observed increased cellulose and lignin contents in CH during the October–January harvest season.

Following the second-stage processing methodology (Figure 3 and Figure 5) and as reported in Section 3, fresh CH was ground and sun-dried. Table 3 compares the composition of sun-dried CH, an important biomass output readily generated in cacao farms and used as an energy source in cacao-producing countries. Moisture content in sun-dried CH affects the combustion efficiency of the material. For sun-dried CH moisture contents in the literature, reports range from 6 to 16 wt %, depending on drying time, ambient temperature, relative humidity conditions, and perhaps the cacao plant variety. However, the literature reports normally do not include information regarding the cacao clone or the compositional changes over time. We found average moisture contents of 9.87 ± 1.40 wt % in sun-dried CH (CCN51). Likewise, ash content is an important parameter to estimate biomass behavior when burned, and as an input in models to describe ash transport and deposition during combustion. High ash contents also affect the production of granulated fuel. The average ash content in sun-dried CH (CCN 51) corresponds to 10.61 ± 1.39 wt %, a value within the ranges of 6 and 13 wt % reported for CH in the literature (Table 3). The lowest ash content in Table 3 (6.7 ± 0.2 wt %) was reported for CH samples from Bahia, Brazil (clone not specified) [53]. The average crude fat content in sun-dried CH (1.61 ± 0.86 wt %) lies within the values of 0.6 and 2.24 wt % reported by various authors, as shown in Table 3. The amount of protein of 5.90 ± 0.91 wt % in CH (Colombian CCN 51) is also within the range of published values but is lower than the 8.6 ± 0.9 wt % and 9.1 ± 1.7 wt % reported in other studies [52,53]. Regarding the average crude fiber, the results of this work lie within the values reported by other researchers, as seen in Table 3.

The proximal analysis for sun-dried CH belonging to individual cacao loads shows no significant seasonal variations, as seen in Figure 6 (Appendix A). Only in some exceptional cases is there a significant change. For instance, in load 12 (January), the moisture percentage was low (6.62 wt %) and the protein content was high (8.08 wt %) compared to the average of the values. Likewise, load 2 (March) showed a low fat content (0.57 wt %) and a high percentage of crude fiber (37.92 wt %). Sun-dried CH calorific values ranged from 13.22 (Mar) to 14.62 (Oct), with an average of 13.69 ± 0.43 MJ kg ^−1^ (Appendix A, Figure 7). The direct combustion of sun-dried CH is common for energy generation in cacao-producing countries. Compared to other residual biomass sources, such as soybean cake, rapeseed, cotton cake, potato peel, apricot bagasse, and peach bagasse, with values from 15.41 to 19.52 MJ kg ^−1^ [57], the calorific value of sun-dried CH is slightly lower, with an average value of 13.69 ± 0.43 MJ kg ^−1^.

According to the data in Table 1 and Table 2 and Figure 5, CH can be cataloged as a lignocellulosic output and, as such, can be processed via physical, chemical, or biological methods to produce added-value byproducts. For instance, the enzymatic or chemical hydrolysis of cellulose in CH can produce sugars, which, in turn, could be used to obtain biofuels such as ethanol, hydrogen, and biobutanol, as well as organic acids through fermentation [58]. Likewise, cellulose could be used for advanced materials production, nanocrystals, and nanofibers. The hemicellulose in CH could be used as a source of probiotics since the main components of this fraction include heteropolymers like xylan, glucuronoxylan, arabinoxylan, xyloglucans, and galactomannans [53,59]. A recent review discussed low- and high-value applications for CH. Among the former, the authors included fertilizer, soil amendments, animal feed, activated carbon, and soap, while the latter involve paper-making, biofuel, dietary fiber, and antioxidant isolation [60].

#### 2.2.2. Cacao Beans (CBs)

CBs are the seeds of the *Theobroma cacao* tree and are the main product of the cacao crop. Table 4 shows the average proximate analysis for dried and fermented cacao beans (CB) from cacao fruit samples of various origins. Colombian CCN51 dried and fermented CBs contain mainly fat (54.41 ± 0.92 wt %), carbohydrates (27.02 ± 0.98 wt %), and protein (12.10 ± 0.44 wt %); these values are within the ranges reported for Ghanaian and Ecuadorian CBs (Table 4). The protein content in Colombian CCN-51 CBs is within the ranges of 10–15 wt % reported for CBs from Venezuela (Trinitarian variety), Ecuador, Ghana (hybrid clones), Peru, Madagascar, Dominican Republic, and the Ivory Coast [61,62,63,64]. The fat content (54.41 ± 0.49) in Colombian CBs is similar to reports of 56–58 wt % for fermented and dried CBs (Forastero variety) in West Africa [63] and is higher than the values of 43.46 ± 0.25 and 41.93 ± 0.13 wt % found in the Ecuadorian and Ghanaian CB samples [61,62]. The carbohydrate content in Colombian CBs (27.02 ± 0.98 wt %) is similar to the reports of 23.1 ± 0.54 wt % in Ghanaian Forastero [63] and lower than the values of 33.79 ± 0.24 and 36.58 ± 0.15 wt % reported for the Ecuadorian and Ghanaian CB samples (clone not specified) [62], as seen in Table 4.

Cacao beans are rich in antioxidants and are, thus, considered a functional food. Many scientific studies demonstrate the link between improved cardiovascular health and the consumption of antioxidant-rich chocolate [65]. Antioxidants in CBs exist mainly as polyphenols (flavonoids), which are responsible for the characteristic bitter taste of raw cacao seeds. The total polyphenols in Colombian CBs (47.31 ± 8.03 mg GAE/g) are in the low end of the range, according to a review of antioxidant content in CBs from various origins. For instance, the polyphenols content in Ecuadorian CCN51 CBs ranged from 84 mg GAE/g to 91 mg/g (catechin equivalents), while the Criollo clones from the Dominican Republic and Peru exhibited lower polyphenol contents of 40 and 50 mg GAE/g [66]. However, the polyphenol content in Colombian CBs was within the reported values of 34 to 60 mg/g, for a series of CB samples from the Ivory Coast (Forastero variety), Colombia (Amazon variety), Equatorial Guinea (Amazon Forastero variety), Ecuador (Amazon-Trinitario-Canelo, Amazon hybrid), Venezuela (Trinitario variety), Peru (Criollo variety), and the Dominican Republic (Criollo variety) [67].

Figure 8 and Appendix A show the seasonal variations in CB composition. There are no significant differences in moisture (2.94 % wt–4.74 % wt), ash content (2.40 wt %–2.93 wt %), fat (52.77 wt %–55.36 wt %), protein (11.34 wt %–12.67 wt %), total carbohydrates (22.39 wt %–26.93 wt %), and calorific value (26.51 MJ kg^−1^–27.42 MJ kg^−1^) for the various CB loads studied. However, parameters such as crude fiber and total polyphenols (Appendix A) in CBs exhibited significant changes over time. For instance, the total polyphenol content reached maxima values during the harvest seasons, with contents ranging from 49.97 to 61.68 mg GAE/g from January to April, and 52.12 to 57.58 mg GAE/g in November and December. The total polyphenols in CCN51 CBs reached minima values from May to September (36.21 to 42.52 mg GAE/g).

We believe that compositional information is fundamental to making informed decisions in cacao crop management. For CBs, for instance, knowing when the beans reach maximum antioxidant values could determine the fate of the product, such as antioxidant extraction vs. roasting, allowing producers to increase their profitability and product portfolio.

#### 2.2.3. Cacao Placenta (CP)

The CP is a fibrous supportive tissue that holds the CBs together inside the pod. According to Appendix A, fresh CP corresponds to 2.58 ± 0.22% of the total fruit weight. The material mainly contains water; after drying, the total solid content in CP is 0.57 ± 0.05 wt %. Perhaps, due to its scarceness, CP has received little attention in the scientific literature. Table 5 shows an average CP moisture content of 78.36 wt % for CCN51 fruit, as in reports of 80.55 ± 0.42 wt % and 80.925 ± 1.06 wt % for Côte d’Ivoire cacao fruit (Forastero variety) [68] and Ecuadorian cacao fruit (CCN51 clone) [69]. We found low ash percentages in CP (0.54 wt %), in contrast with the 9.34 wt % reported for Côte d’Ivoire cacao fruit [68]. The protein content (1.69 wt %) is within the values reported by other authors. The total nitrogen in CP (0.27 wt %) is lower than 1.005 ± 0.134 wt %, as reported by the authors of [69].

Lately, CP has attracted some attention as a source of nutrients, antioxidants, and bioactive compounds. For instance, one study has found that fermentation increased the antioxidant activity of CP, while the levels of sugars, tannins, and flavonoids decreased [69]; likewise, the nutritional value of dehydrated CP makes it an ingredient in nutritional bars [31] or for the production of an alcoholic drink and nectar made from CP from Ecuadorian cacao fruit samples (CCN51 clone) [30]. In the field of animal nutrition, previous studies reported the use of dried CP from Ecuadorian cacao in chicken feed formulations [70] and quantities of CP flour (CPF) for pig feeding [71].

#### 2.2.4. Cacao Mucilage Exudate (CME)

Cacao mucilage exudate or sweatings (CME), a sweet liquid that seeps from the white pulp surrounding the cacao bean, is typically lost during the cacao bean fermentation process. CME comprises up to 4.13 ± 0.80 wt % of the cacao fruit’s biomass (see Appendix A) and is mainly composed of water and carbohydrates. Table 6 compares the average proximate analysis for CME from cacao fruit samples of various origins and from this work. The average moisture content of Colombian CCN51 CME (82.68 ± 6.09 wt %) is similar to the values of 85.86 ± 0.09 wt % reported for samples from Bahia, Brazil (clone not specified) [72], and 82.5 % and 80.5 % for the Ecuadorian samples (Nacional and CCN-51 clones) [73]. The protein content in the Colombian CCN51 CME samples (0.31 ± 0.09 wt %) is lower than (1.2 ± 0.49 wt %) [69], (5.41 wt %) from the Ecuador cacao fruit (CCN-51 clone) [74], and (5.56 ± 0.1 wt % and 5.47 ± 0.12 wt %) from the Taura and Cone Ecuadorian samples [11]. The carbohydrate content in CCN51 CME has an average value of 17.18 ± 6.44 wt %, lower than the values of 68.35 ± 0.16 and 67.99 ± 0.14 for Cone and Taura Ecuadorian samples [11]. CME contains small amounts of polyphenols (0.55 ± 0.45 mg gallic acid/g), a characteristic that has not previously been reported. Appendix A includes the detailed proximate analysis of Colombian CCN51 CME samples collected over one year.

Figure 9 and Appendix A show the seasonal variations in CME composition. There are significant differences in moisture, with higher values observed toward the end of the year from August to January (88.6 wt %) and lower values during February and March (71.5 wt %). The ash (0.27 wt % to 0.61 wt %) and protein contents (0.17 wt % to 0.46 wt %) in CME are low. In contrast, the total carbohydrate content in CME changes dramatically during the year, with a minimum of 10 wt % during August and January and a maximum of 29 wt % in March. Interestingly, the first three loads of the year showed the highest total carbohydrate content and calorific values, and the lowest moisture content of all samples.

CME’s high sugar content hints at a valuable biomass output, with potential uses as human food or as a carbon source in industrial processes. The average sugar values in CME from CCN51 cacao fruit correspond to 75.54 ± 9.54 g L^−1^ for fructose, 67.15 ± 8.36 g L^−1^ for glucose, and 14.08 ± 8.89 g L^−1^ for saccharose. The glucose and fructose contents are higher than the values of 45.8 ± 1.2 g L^−1^ and 32.5 ± 0.3 g L^−1^ [72]; however, the glucose content is much lower than the 214.2 ± 6.2 g L^−1^ reported for South Cote D’Ivoire samples (clone not specified) [25].

There are significant differences between individual sugars in CME from different cacao fruit loads, as seen in Figure 10. The saccharose content ranged from 6.84 g L^−1^ to 33.7 g L^−1^, while the glucose and fructose contents stretched from 54.3 g L^−1^ to 80.89 g L^−1^ and from 59.3 g L^−1^ to 84.43 g L^−1^, respectively. The content of the most complex sugar in CME (saccharose) increases, at the expense of a decreased content of glucose and fructose during the main harvest (October–December), which is also the dry season in the region. However, from October to December, we also registered the lowest total carbohydrate content in CME. Sugar concentration in CME depends on many factors, such as fruit ripeness, rainfall, crop conditions, the time of year, and cacao variety, among others. Sugar accumulation can be enhanced by more prolonged exposure of the fruit to the sun [75]. A Pearson’s correlation coefficient (r) of 0.63 suggests a direct correlation between the sugar content in CME and rainfall in the area. The CME extracted from cacao fruit loads processed during November, December, and January exhibit the lowest sugar content, matching up with the dry season (low rainfall). On the other hand, during the rainy season (March to October—an abnormally long rainy season), we observed the highest total sugar content. Appendix A contains detailed seasonal compositional information for CME.

CME also contains trace minerals in the form of calcium, sodium, potassium, and aluminum. These micronutrients are relevant nowadays because of their proven benefits to the human body. Sodium, for instance, an essential mineral that allows maintaining the water balance in the body and adequate blood pressure, has an average concentration of 1.71 ± 0.078 ppm in CME. In contrast, cacao beans exhibit higher Na contents, ranging from 3 to 32 ppm for samples from various origins [76]. Potassium, which is also fundamental for water balance and cardiovascular health [77], has an average concentration of 2413.77 ± 194.89 ppm in CME. The high potassium content in CME makes it a natural source of the mineral, rivaling traditional high-potassium foods such as potatoes, bananas, apricots, tomatoes, carrots, passion fruit, and many others. In contrast, reports dealing with the multi-elemental analysis of cacao beans from worldwide samples show K contents of around 10–13 ppm, suggesting a selective accumulation of this nutrient in the mucilage surrounding the beans [76]. Interestingly, Na and K contents increased in processed cacao bean byproducts, such as cacao mass and cocoa [78].

The literature reports several uses for CME. For instance, the use of CME for making marmalade, ending up with a product with similar organoleptic properties as apricot marmalade and a nutritional value comparable with some tropical fruits [25], following jelly formulations from CME, extracted from national varieties and CCN-51 [73]. Likewise, jam was made with 67.14° Brix using CME, cacao placenta, and added cane sugar [25]. A cacao beverage containing CME and a liqueur was extracted from Trinidad and Tobago cacao beans (hybrid clone) [26]. CME’s uses are also reported in patents. For instance, one application suggested a method for obtaining a syrup from CME and cacao mucilage from CBs that was suitable for food preparations as a flavoring or texturizing agent [27].

The high sugar content in CME makes it an ideal carbon source for biotechnological applications. For instance, one study used CME and *Saccharomyces cerevisiae* to produce a fermented drink with 22.4% of ethanol [32]. In the same way, the production of ethanol from CME fermentation with *Saccharomyces cerevisiae*, with a maximum yield of 0.073 g g^−1^ and volumetric productivity of 0.168 g L^−1^ h^−1^ was achieved [72]. In another work, the authors reached a maximum concentration of 13.8 g L^−1^ of ethanol, the production of 0.5 g of ethanol g glucose ^−1^, and a productivity of 0.25 g L ^−1^ h^−1^ of ethanol from CME fermented with *Pichia kudriavzevii* [33]. Elsewhere, CME from cacao fruit was sourced from the Selemadeg Barat District, Bali Province (clone not specified) to produce vinegar [79]. The authors found that a single-phase fermentation, plus additional alcohol, resulted in a high acetic and propionic acid product. Finally, more recently, our group demonstrated the feasibility of using CME for bacterial cellulose (BC) production [34]. We measured maximum bacterial cellulose yields and achieved production rates of 15.57 g L^−1^ and 0.041 g L^−1^ h^−1^, which were replicated at laboratory and pilot-plant scales. This observation suggests an industrial scenario, wherein BC production from CME is a real possibility.

## 3. Materials and Experimental Methods

### 3.1. Area of Study

Ripe cacao fruits were harvested from trees of the CCN51 clone ranging from 6 to 10 years of age. The cacao plantation was situated in the rural area of San Vicente de Chucurí (latitude: 6°52′59′′ N, longitude: 73°25′1′′ W) at the heart of the main cacao-producing region in the country (Santander—Colombia). In 2018, San Vicente de Chucurí contributed 28 wt % to Santander’s cacao production [80].

### 3.2. Cacao Fruit Collection and Treatment

Ripe cacao fruit loads, of approximately one tonne each (1000 kg), were brought to the lab monthly (February–January of 2018–2019) for a total number of twelve loads (12,000 kg). For the cacao fruit compositional and mass balance analyses, we devised a two-stage procedure. The first stage involved opening the cacao fruit and separating the fresh cacao beans (CBs) and the placenta (CP) from the husk (CH), as seen in Figure 3. This first process represents the traditional approach to cacao fruit usage. In the second stage, the CP and the CH were ground, dried, and stored for further compositional analysis (Figure 3). After CB extraction, the CH and CP were ground down to a particle size of 0.1–2 cm, using a TRAPP TRF 300 mill (Trapp, Brazil), and were then sun-dried for 5–6 days. The CH was turned over every 4 h to facilitate water evaporation.

Fresh CBs were placed in a double-wall stainless-steel container fitted with an inner mesh that allowed the collection of the cacao mucilage exudate (CME) designed and built by our research group. The same container was used to perform the fermentation process for seven days. The CME and CBs were stored for further analysis. All materials (wet and dry) were weighed to determine the mass balances. Table 7 shows the equations used to determine the percentages in terms of wet and dry weight for the cacao fruit byproducts, as reported throughout the manuscript.

The percentage weight and compositional information reported throughout the manuscript correspond to the average values and standard deviations from twelve measurements. Statistics analysis was performed, using the average and standard deviation functions, in an Excel spreadsheet. The Pearson coefficient was also calculated, using the correlation function, in an Excel spreadsheet.

Table 7 illustrates the equations used to determine the percentages in wet and dry weight for the cacao fruit byproducts.

### 3.3. Compositional Analyses

Table 8 shows the analysis and reference methods used to characterize the components isolated from the cacao fruit: cacao husk (CH), cacao placenta (CP), cacao mucilage exudate (CME), and cacao beans (CBs).

## 4. Conclusions

We identified five distinctive biomass outputs from cacao fruit: cacao husk (CH), cacao beans (CBs), cacao placenta (CP), and cacao mucilage exudate (CME). CH, CBs, and CP are solid lignocellulosic outputs that comprise, in terms of dry weight, 8.92 ± 0.90 wt %, 8.87 ± 0.52 wt %, and 0.57 ± 0.05 wt % of the cacao fruit weight, respectively. Moisture, on the other hand, constitutes most of the biomass weight (71.6 ± 2.29 wt %). Cellulose and lignin contents in CH are time-dependent, reaching maximum values during the crop’s primary harvest season (October–January).Dried CH is mostly used as an energy source in the cacao-producing regions of the world. We found no significant changes in CH calorific values during the crops’ yearly cycle, with an average of 13.69 ± 0.43 MJ kg^−1^. This value is similar to the calorific value content in other residual biomass outputs, such as rice straw/husk, soybean cake, potato peels, rapeseed, sugarcane bagasse, and cotton cake.As a lignocellulosic output, CH can potentially be processed via physical, chemical, or biological methods to produce added-value byproducts, such as simple sugars, ethanol, hydrogen, biobutanol, and volatile organic acids, among others. Likewise, the structural biopolymers in CH, such as cellulose, can be the precursors of high-performance materials, such as nanocrystals and nanofibers. The hemicelluloses in CH, which are rich in heteropolymers like xylan, glucuronoxylan, arabinoxylan, xyloglucans, and galactomannans, could become a potential source of probiotics such as xylo-oligosaccharides.CB contains carbohydrates, fats, protein, ash, and phenolic compounds. The contents of these materials in Colombian CCN51 CBs do not change significantly during the yearly crop cycle and are within the range of CBs from different geographical sources. Interestingly, the total polyphenol content in CBs is time-dependent, reaching maxima values during the harvest seasons. For instance, from January to April, CBs exhibit 49.97 to 61.68 mg GAE/g, and, from November to December, 52.12 to 57.58 mg GAE/g. The total polyphenols in CCN51 CBs reach minimum values of 36.21 to 42.52 mg GAE/g from May to September.Cacao mucilage exudate (CME) is a liquid biomass output that is equivalent to 4.13 ± 0.80 wt % of the cacao fruit. CME is rich in simple sugars (glucose, fructose, and saccharose) and minerals (K), with an average of 20 wt % of total carbohydrates. Interestingly, the total carbohydrate content in CME changes dramatically during the year, with a minimum of 10 wt % from August to January and a maximum of 29 wt % in March. Likewise, we observed a positive correlation between sugar content in CME and rainfall, with the highest sugar content in CME being measured during April when rainfall was at its highest.CME uses include fermentation to produce alcohol and concentration to produce syrups and jams. However, the high nutrient content of CME makes it an ideal culture media for biotechnological applications, particularly biopolymer production, as demonstrated recently by our group.

## Figures and Tables

**Figure 1 molecules-27-03717-f001:**
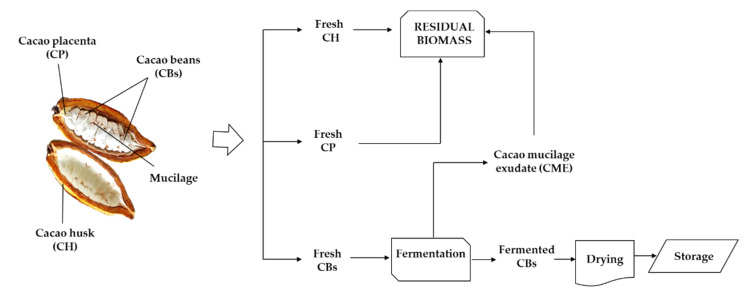
Traditional scheme for cacao fruit processing in tropical cacao-producing countries.

**Figure 2 molecules-27-03717-f002:**
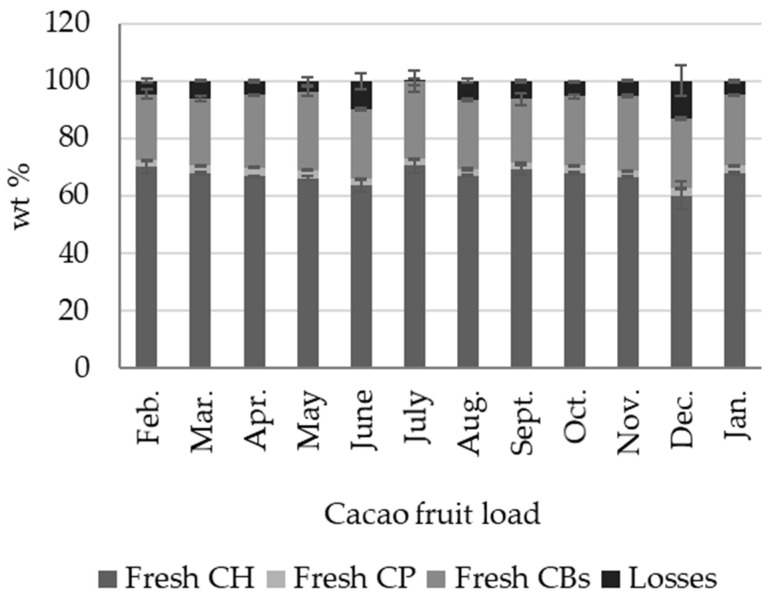
Seasonal variations in the percentage weight of fresh cacao fruit biomass outputs, cacao husk (CH), cacao placenta (CP), and cacao beans (CBs), derived from the traditional approach of cacao fruit processing (first stage, Figure 1).

**Figure 3 molecules-27-03717-f003:**
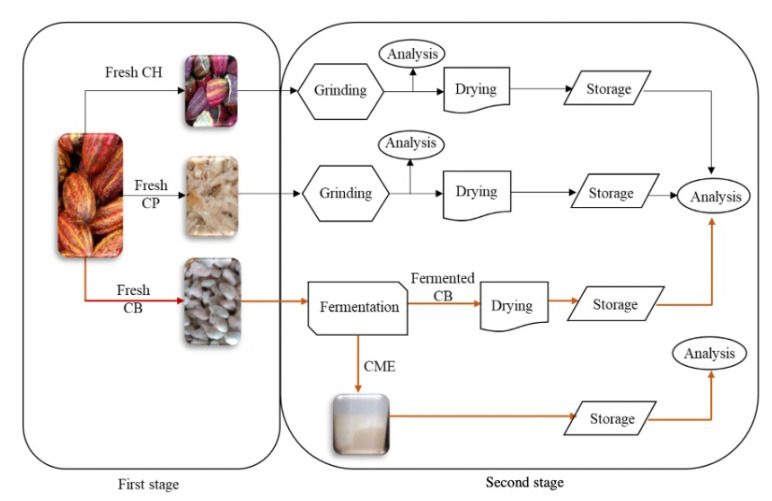
General scheme of cacao-fruit processing for this work.

**Figure 4 molecules-27-03717-f004:**
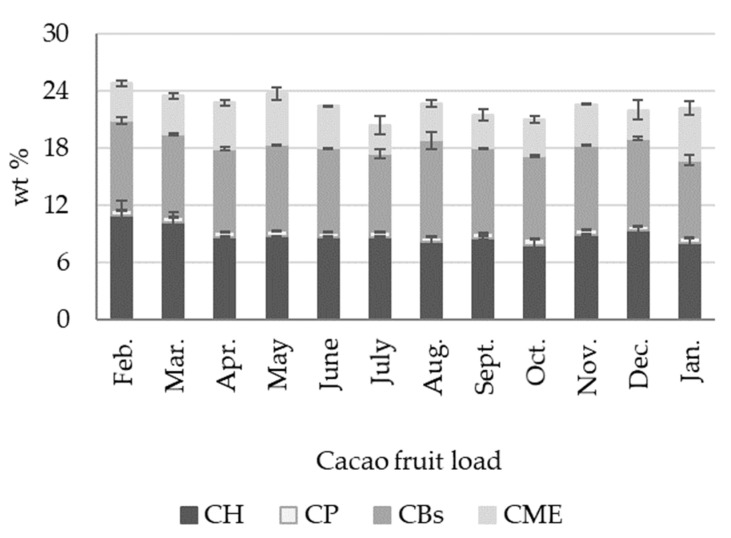
Seasonal variations in the percentage weight of cacao fruit biomass outputs, dried cacao husk (CH), dried cacao placenta (CP), dried cacao beans (CBs), and liquid cacao mucilage exudate (CME) derived from the second stage of cacao fruit processing (Figure 3). Note: the remaining percentage corresponds to moisture.

**Figure 5 molecules-27-03717-f005:**
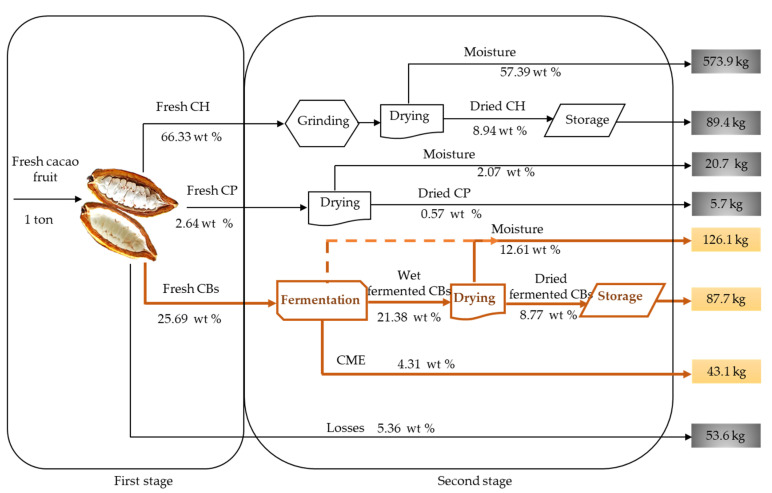
Mass balance for 1 tonne of fresh cacao fruit during the first and second stages of processing (load: Nov.).

**Figure 6 molecules-27-03717-f006:**
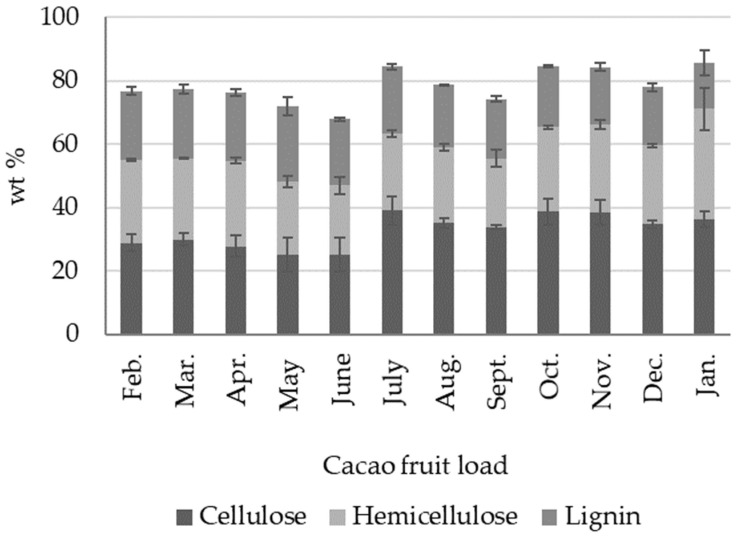
Seasonal variations in structural carbohydrates (cellulose and hemicellulose) and lignin content in CH as fractions of the total solids in fresh CH, from Colombian CCN51 cacao fruit.

**Figure 7 molecules-27-03717-f007:**
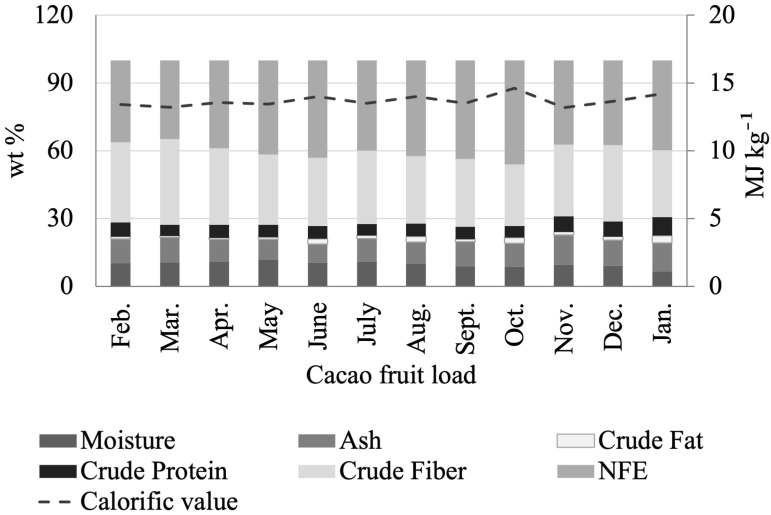
Seasonal variations in the proximate analysis of sun-dried cacao husk (CH) from Colombian CCN51 cacao fruit.

**Figure 8 molecules-27-03717-f008:**
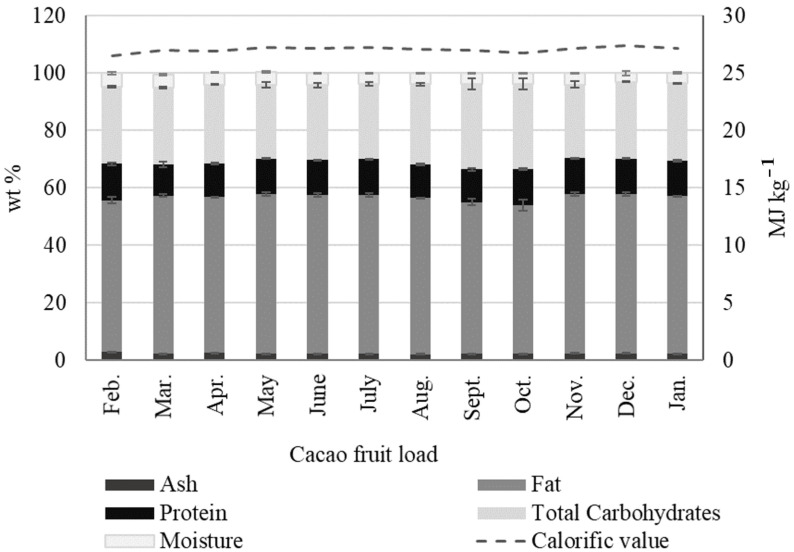
Seasonal variations in the proximate analysis of fermented and dried cacao beans (CB) from Colombian CCN51 cacao fruit.

**Figure 9 molecules-27-03717-f009:**
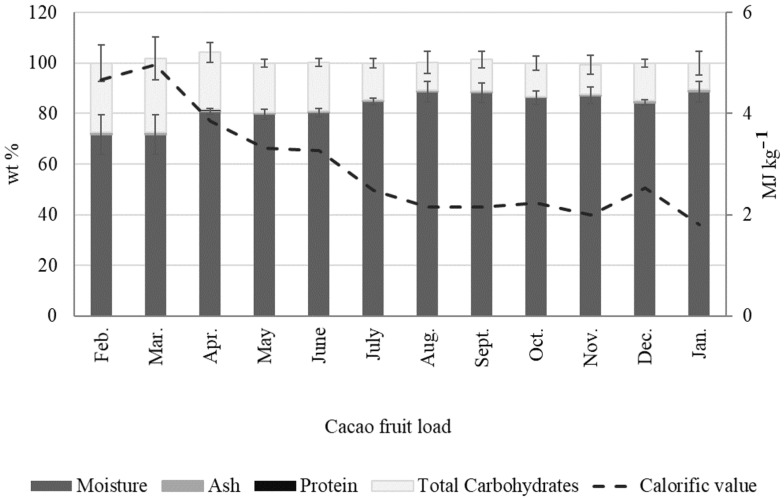
Seasonal variations in the proximate analysis of CME from Colombian CCN51 cacao fruit.

**Figure 10 molecules-27-03717-f010:**
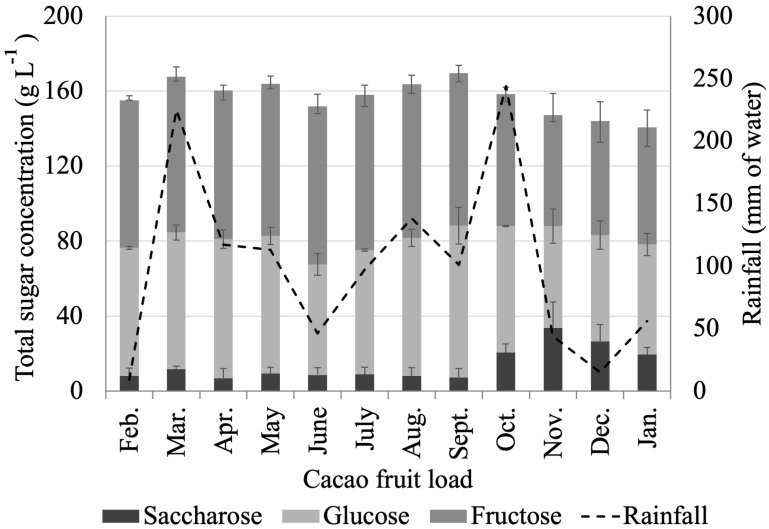
Seasonal variations in CME sugar content from Colombian CCN51 cacao fruit.

**Table 1 molecules-27-03717-t001:** Average proximate analysis of fresh cacao husk (CH) from clone CCN51 cacao fruit.

Component.		Percentage by Weight (wt %)
Moisture		84.62 ± 1.97
Total solids		15.38 ± 1.97
	Crude fat *	0.29 ± 0.08
	Crude protein *	5.44 ± 0.80
	Crude fiber *	30.93 ± 2.03
	Ash *	10.65 ± 1.62
	Total nitrogen *	0.88 ± 0.13
	Nitrogen-free extract (NFE) *	52.69 ± 2.50

* As a percentage of the total solids.

**Table 2 molecules-27-03717-t002:** Structural carbohydrates and lignin in cacao husk (CH) from cacao fruit samples of various origins and this work.

Cellulose	Hemicellulose	Lignin	References
(wt %)			
31.7 ± 0.1	27.0 ± 0.1	21.7 ± 0.1	[10]
30.79	21.09	25.55	[48]
18.42	10.04	12.06	[49]
35	10	14.6	[50]
30.41 ± 0.20	11.97 ± 3.17	33.96 ± 1.9	[47]
35.4 ± 0.33	37 ± 0.5	14.7 ± 0.35	[42]
41.92 ± 0.09	35.26 ± 0.05	0.95 ± 0.04	[51]
26.15 ± 3	12.7 ± 56	21.16 ± 2.6	[52]
25.64 ± 3.49	19.96 ± 2.42	32.73 ± 5.15	This work *

* As a percentage of the average total solids value from Table 3.

**Table 3 molecules-27-03717-t003:** Average proximate analysis for sun-dried CH from cacao fruit samples of various origins.

Moisture	Ash	Crude Fat	Crude Protein	Crude Fiber	Reference
wt %					
12.5	12.3	VNR	VNR	VNR	[12]
VNR	11.44 ± 0.41	0.93 ± 0.34	2.42 ± 0.37	VNR	[39]
10.5	9	1.5	2.1	VNR	[54]
VNR	VNR	2.5	8.4	32.3	[40]
13	13	0.6	8	50	[23]
VNR	9.1–10.1	VNR	5.9–7.6	22.6–32.5	[50]
VNR	VNR	1.2–10	5.9–9.1	22.6–35.7	[36]
10.04 ± 0.3	12.67 ± 0.14	VNR	VNR	33.6 ± 0.15	[55]
6.72 ± 0.17	8.32 ± 0.7	2.24 ± 0.1	4.22 ± 0.07	VNR	[11]
8.5 ± 0.6	6.7 ± 0.2	1.5 ± 0.13	8.6 ± 0.9	36.6 ± 0.01	[53]
16.1	13.5	VNR	VNR	VNR	[56]
VNR	9.07 ± 0.04	VNR	9.1 ± 1.7	35.7 ± 0.9	[52]
9.86 ± 0.75	10.61 ± 1.39	1.61 ± 0.86	5.90 ± 0.91	31.91 ± 2.98	This work

VNR: Value not reported.

**Table 4 molecules-27-03717-t004:** Average proximate analysis for dried and fermented cacao beans (CB) from cacao fruit samples of various origins.

Moisture	Ash	Crude Fiber	Crude Protein	Crude Fat	Carbohydrates	Total Polyphenols (mg Gallic Acid/g)	Reference
wt %							
4.3 ± 0.09	2.3 ± 0.04	VNR	18.2 ± 0.13	52.2 ± 0.1	23.1 ± 0.54	VNR	[63]
6.22 ± 0.2	2.84 ± 0.04	VNR	12.25 ± 0.16	42.7 ± 0.6	VNR	VNR	[61]
5.95 ± 0.04	4.03 ± 0.01	VNR	12.79 ± 0.03	VNR	33.78 ± 0.02	VNR	[62] *
5.11 ± 0.01	3.56 ± 0.02	VNR	12.82 ± 0.01	VNR	36.58 ± 0.01	VNR	[62] +
3.96 ± 0.50	2.51 ± 0.17	3.20 ± 0.90	12.10 ± 0.44	54.41 ± 0.92	24.00 ± 2.24	47.31 ± 8.03	This work

VNR: Value not reported. * Ecuador, + Ghana.

**Table 5 molecules-27-03717-t005:** Average proximate analysis for fresh CP from cacao fruit samples of various origins.

Moisture	Ash	Protein	Total Nitrogen	Reference
wt %				
80.55 ± 0.42	9.34 ± 0.89	5.12 ± 0.02	VNR	[68]
80.925 ± 1.067	5.560 ± 0.424	VNR	1.005 ± 0.134	[69]
VNR	1.28 ± 0.051	1.38 ± 0.028	VNR	[30]
78.36	0.54	1.69	0.27	This work

VNR: Value not reported.

**Table 6 molecules-27-03717-t006:** Average proximate analysis for CME from cacao fruit samples of various origins and this work.

Moisture	Ash	Protein	Total Carbohydrates	Total Polyphenols (mg Gallic Acid/g)	Reference
wt %					
85.86 ± 0.09	0.59 ± 0.15	1.20 ± 0.49	11.80 ± 0.09	VNR	[72]
77.34 ^a^	2.91	5.41	VNR	VNR	[73]
82.5 ^b^	VNR	0.87	VNR	VNR	[73]
80.5	VNR	0.38	VNR	VNR	[74]
VNR ^c^	7.51 ± 0.14	5.47 ± 0.12	68.35 ± 0.16	VNR	[11]
VNR ^d^	7.68 ± 0.18	5.56 ± 0.10	67.99 ± 0.14	VNR	[11]
82.67 ± 6.09	0.45 ± 0.01	0.31 ± 0.09	17.18 ± 6.44	0.55 ± 0.45	This work

VNR: Value not reported. ^a^ Nacional, ^b^ CCN-51, ^c^ Cone, ^d^ Taura.

**Table 7 molecules-27-03717-t007:** Equations used to determine the percentages by weight of cacao fruit outputs.

Percentages	Equation	Definitions
Wet weight	$\mathrm{CH}\;\mathrm{wt}\;\%=\frac{M_{\mathrm{FCH}}}{M_{\mathrm{FCF}}}$	M_FCH_ mass of fresh CH M_FCF_ mass of fresh cacao fruit
	$\mathrm{CP}\;\mathrm{wt}\;\%=\frac{M_{\mathrm{FCP}}}{M_{\mathrm{FCF}}}$	M_FCP_ mass of fresh CP
	$\mathrm{CB}\;\mathrm{wt}\;\%=\frac{M_{\mathrm{FCBs}}}{M_{\mathrm{FCF}}}$	M_FCBs_ mass of fresh CBs
	Fermented CBs wt %= CBs wt %−CME wt %	
Dry weight	$\mathrm{Dried}\;\mathrm{CH}\;\mathrm{wt}\;\%=\frac{M_{\mathrm{DCH}}}{M_{\mathrm{FCF}}}$	M_DCH_ mass of dried CH
	$\mathrm{Dried}\;\mathrm{CP}\;\mathrm{wt}\;\%=\frac{M_{\mathrm{DCP}}}{M_{\mathrm{FCF}}}$	M_DCP_ mass of dried CP
	$\mathrm{Dried}\;\mathrm{fermented}\;\mathrm{CBs}\;\mathrm{wt}\;\%=\frac{M_{\mathrm{DFCBs}}}{M_{\mathrm{FCF}}}$	M_DFCBs_ mass of dried fermented CBs
Moisture content (MC)	${\mathrm{CH}}_{\mathrm{MC}}\;\%=\frac{\left(M_{\mathrm{FCH}}-M_{\mathrm{DCH}}\right)}{M_{\mathrm{FCH}}}$	M_FCH_ mass of fresh CH M_DCH_ mass of dried CH
	${\mathrm{CP}}_{\mathrm{MC}}\;\%=\frac{\left(M_{\mathrm{FCP}}-M_{\mathrm{DCP}}\right)}{M_{\mathrm{FCP}}}$	M_FCP_ mass of fresh CP M_DCP_ mass of dried CP
	${\mathrm{CB}}_{\mathrm{MC}}\;\%=\frac{\left(M_{\mathrm{FCBs}}-M_{\mathrm{DCB}}\right)}{M_{\mathrm{FCB}}}\;{\mathrm{CME}}_{\mathrm{MC}}\;\%=\frac{\left(M_{\mathrm{FCME}}-M_{\mathrm{DCME}}\right)}{M_{\mathrm{FCME}}}$	M_FCBs_ mass of fresh CBs M_DCP_ mass of dried CP M_FCME_ mass of fresh CME M_DCME_ mass of dried CME

**Table 8 molecules-27-03717-t008:** Chemical characterization of the various cacao fruit components.

Analysis	Reference Method	CH	CP	CME	CBs
Moisture	AOAC 931.04 [81]	X		X	X
	AOAC 925.10 [82]		X		
Ash	AOAC 972.15 [83]	X		X	X
	AOAC7,009/84-94205/90		X		
Protein	AOAC 970.22 [84]	X		X	X
	AOAC 2001.11 [85]		X		
Crude fiber	AOAC 930.20a [86]	X	X		X
Fat	AOAC 920.75a [87]	X	X		X
Total carbohydrates	By difference: 100—(% Ash)—(% Total Fat)—(% Moisture)—(% Protein)—(% Fiber)	X	X		X
Calorific value	By equation: (% T Carbohydrate × 4 Kcal/g) + (% Protein × 4 Kcal/g) + (% Total Fat × 9 Kcal/g)	X	X		X
Total polyphenols	Standard Methods 5530 B, D [88]			X	X
Glucose	AOAC 925.36 [89]			X	
Fructose	AOAC 925.36			X	
Sucrose	AOAC 925.36			X	
Total soluble solids °Brix	AOAC 931.12 [90]			X	
pH 24.2 (°C)	AOAC 960.19 [91]			X	
Calcium	AOAC 985.35 [92]			X	
Potassium	AOAC 985.35			X	
Sodium	AOAC 985.35			X	
Aluminum	Emission mode			X	
Total nitrogen	Standard Methods 4500 N [93]	X	X	X	
Total solids	Standard Methods 2540B [94]	X			
Holocellulose	Jayme-Wise Method [95]	X			
Cellulose	Kurschner and Hoffer Method [96]	X			
Hemicellulose	By difference: % Holocellulose—% Cellulose	X			
Lignin	Klason Method [97]	X			

## Data Availability

The data presented in this study are available in supplementary material.

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
