# Peer review of "Mass Balance and Compositional Analysis of Biomass Outputs from Cacao Fruits"

_molecules, 2022, doi:10.3390/molecules27123717_

Round 1

Reviewer 1 Report

The manuscript entitled “Mass balance and compositional analysis of biomass outputs from cacao fruits” aims to present cacao fruits composition and mass-balance dynamics by analyzing representative samples collected monthly for one year. The manuscript is well prepared. The introduction is informative, the results are comprehensible, and the results support the conclusions. Still, I think that discussion can be improved. Also, the potential for cacao biomass application is enormous, so it should be at least demonstrated. Without the application, I don’t find the presented results meaningful.

The authors suggest that the residues in the cacao fruits processing could be used as raw materials to feed processes to produce value-added products to strengthen the cacao value chain in producers’ countries. In my opinion, the authors should try to find some application for this biomass. Maybe try to pyrolyze it and use it as an adsorbent for some organic pollutants removal.

Reviewer 2 Report

abstract

line 20 and 23- this lines contains the same information, please reduce repetitive information

The authors focused only on the composition of cocoa pod and reported the moisture content of the individual components. This information should be condensed and given in one sentence. However, there is no other compositional information and the main conclusion of the study is missing. In view of the above, the abstract should be corrected/rewritten.

Introduction

The authors have written the introduction to the topic well. They presented well the validity of the subject matter undertaken in the manuscript and justified conducting the research presented. 

Figure 1 –please provide figure with better resolution quality

line 51-52 –needs citations of literature to support the words; the best will be reviews

line 114-115- given values are your own values that comes from current research so please delete it here. Don’t give values from research in introduction. The text may remain but without values.

Results and discussion

line 140- whether the authors did statistics for the data presented? there is no description of the statistical methods and software used for statistics in the methodology of the manuscript. maybe add letters on figure 2 to show homogeneous groups

figure 3- why is the sum of all values do not equal 100%? What makes up 74% of the rest- moisture? please explain this in the figure footnote

line 202-205 –please rewrite these sentences – give better description of other works/studies with which you compare yours

line 343- this sentence needs citation of mentioned review

line 441-442 - have you done any statistical analysis such as Pearson correlation coefficient to prove this statement?

line 450-458 – please discuss the minerals analysis results with minerals content in cocoa beans, because there is close connection of CME with cocoa beans. Here are my recommendations: doi.org/10.1016/j.foodcont.2016.01.013,  doi.org/10.1016/j.lwt.2018.08.030

Materials and experimental methods

line 485 and 489- you repeated twice that cocoa trees were the CCN51 clone, please rewrite to delete the repetitive information

line 486-487 – maybe authors could provide the coordinates of the cocoa plantation?

line 488-489- Did the authors mean to say that cocoa pods were collected when they were considered ripe? please clarify this sentence in manuscript

line 493- February –January of which years?

line 495-496- please describe for readers here what is cacao placenta, and how it differs from cocoa mucilage?

lines 492-493 and 506-508 - it is a repetition of the same information, please check and correct

line 511-512 and table 7 - should be moved to another part of the methodology section

table 7 – the equation for CME moisture is missing

table 8 – 1) according to table 7 moisture of CP was measured  but in table 8 this measurement is not marked; 2) in total carbohydrates equation authors omitted to subtract the % fiber 3) in Calorific value equation authors omitted to count %fiber as 2 kcal/g, and must make a correction for the carbohydrate content 4) add citations at each method in the table

Conclusions

conclusions 1, 2 - these are not conclusions based on the authors' research but merely summaries and generalizations

conclusion 3 - needs to be shortened and should be rewritten in terms of the potential use of cocoa pod constituents (potential outcome of obtaining these values), and not only give values from results section

conclusion 7 - this is rather statement than conclusion, please rewrite it

Round 2

Reviewer 1 Report

Although the authors did not provide the results for the application of the biomass, I see their point and recommend the manuscript for publication in its present form.

Author Response

Bucaramanga-Colombia, May 12 2022

Professor

Makamas Tawatchai

Editor

Molecules

Dear profesor Tawatchi,

We are submitting the answers to the second round of review related to manuscript ID molecules-1695546 “Mass balance and compositional analysis of biomass outputs from cacao fruits”.

Best regards,

  -Marianny Y Combariza

Review report 1

  1. Although the authors did not provide the results for the application of the biomass, I see their point and recommend the manuscript for publication in its present form.

Answer: We thank the reviewer for his/her understanding.  We also acknowledge his/her contribution to improving our manuscript.

Reviewer 2 Report

The authors made a lot of corrections and additions. One more point remained to be added to the manuscript text as a subsection of the methodology:

Authors needs to make a subsection in Materials and experimental methods about statistics done with data (average with standard deviations from twelve measurements, Pearson correlation coefficient) and software used for this purpose.

Author Response

Bucaramanga-Colombia, May 12 2022

Professor

Makamas Tawatchai

Editor

Molecules

Dear profesor Tawatchi,

We submit the answers to the second round of review related to manuscript ID molecules-1695546 “Mass balance and compositional analysis of biomass outputs from cacao fruits”.

Best regards,

  -Marianny Y Combariza

Answers to Review report 2

  1. The authors made a lot of corrections and additions. One more point remained to be added to the subsection of the methodology:

Authors needs to make a subsection in Materials and experimental methods about statistics done with data (average with standard deviations from twelve measurements, Pearson correlation coefficient) and software used for this purpose.

Answer: We thank the reviewer for his/her comment.  We added the following paragraph to the Methods section:

Page 17, Lines 552-555 “The percentage weight and compositional information reported throughout the manuscript correspond to average values and standard deviations from twelve measurements. The statistics analysis was performed using the average and standard deviation functions in an excel spreadsheet. Also, the Pearson coefficient was calculated using a correlation function in an excel spreadsheet.”
